# Behavior Testing in Rodents: Highlighting Potential Confounds Affecting Variability and Reproducibility

**DOI:** 10.3390/brainsci11040522

**Published:** 2021-04-20

**Authors:** Rachel Michelle Saré, Abigail Lemons, Carolyn Beebe Smith

**Affiliations:** Section on Neuroadaptation and Protein Metabolism, National Institute of Mental Health, National Institutes of Health, Department of Health and Human Services, Bethesda, MD 20814, USA; Rachel.Sare@nih.gov (R.M.S.); abigaillemons7@gmail.com (A.L.)

**Keywords:** behavior testing, variability, reproducibility, gene × environment interaction, rodent, translational research

## Abstract

Rodent models of brain disorders including neurodevelopmental, neuropsychiatric, and neurodegenerative diseases are essential for increasing our understanding of underlying pathology and for preclinical testing of potential treatments. Some of the most important outcome measures in such studies are behavioral. Unfortunately, reports from different labs are often conflicting, and preclinical studies in rodent models are not often corroborated in human trials. There are many well-established tests for assessing various behavioral readouts, but subtle aspects can influence measurements. Features such as housing conditions, conditions of testing, and the sex and strain of the animals can all have effects on tests of behavior. In the conduct of behavior testing, it is important to keep these features in mind to ensure the reliability and reproducibility of results. In this review, we highlight factors that we and others have encountered that can influence behavioral measures. Our goal is to increase awareness of factors that can affect behavior in rodents and to emphasize the need for detailed reporting of methods.

## 1. Introduction

Behavior testing is a critical tool in the study of nervous system disorders in rodent models. In our experience, there are numerous influences on behavioral measures, some of which we are likely unaware. Our purpose here is to shed light on some of these factors with the hope that investigators will consider these variables in their study designs and their reports. Most behavioral studies are performed on inbred strains to control for genetic influences, but there remain many additional factors that can make such studies challenging [1]. Variability from unknown sources makes it difficult to reproduce findings from other labs and is a major issue in the translation of findings into knowledge of human conditions. This variability may be one of the reasons that preclinical studies in rodents are often not corroborated in human clinical trials [2,3].

There are subtle procedural differences that can alter results. It is important to consider the many factors that can influence behavior in rodents when planning and reporting behavioral studies. By carefully controlling and transparently reporting methodological considerations, we may be able to reduce variability and enhance reproducibility, thus strengthening preclinical studies.

In this review, we focus on some known factors that have been shown to influence rodent behavior. Some of these have been comprehensively reviewed by others. The majority of the literature we refer to pertains to mice, and some to rats, though there are few studies on other rodents. We do not intend to delve deeply into mechanisms and particulars of influential factors. Our purpose in this review is to bring awareness to the myriad of factors that can affect a rodent’s behavior and to caution investigators to design experiments with these influences in mind. We also emphasize the need for reporting experimental details regarding conditions of testing so that others can try to interpret and replicate results. Additionally, we do not particularly focus on the behavioral domain that is affected, as one domain is likely to influence another. For example, hyperactivity can influence the readout of the standard social behavior test [4], chronic loss of sleep can also influence other behaviors [5], and changes in anxiety often alter many other behaviors. We divide these factors into three categories: Animals, housing conditions, and experiment conditions.

## 2. Animals

### 2.1. Strain

The strain of the mouse is a well-known and important consideration. For example, C57BL/6J mice tend to have better performance on the Morris Water Maze than 129Sv/J mice [6]. Additionally, BALB/cJ mice show higher anxiety-like behavior than C57BL/6J mice [6]. It should also be noted that even substrains of C57Bl/6 mice can be distinguished by different behavioral phenotypes [7]. The strains used in each study should be consistent, as well as explicitly reported (including substrain), for reproducibility.

### 2.2. Sex

Sex differences in behavioral phenotypes have been reported in locomotor activity [8], anxiety-like behavior [8], social behavior [8,9], pain response [10], and certain types of learning and memory [11]. For example, in C57BL/6J and BALB/cJ mice, females were found to be more social than males [8]. In C57BL/6 mice, males performed better on a cued Morris Water Maze task than females [11]. It is important to study both sexes separately to yield complete and accurate results. The importance of this factor is becoming increasingly recognized, and sex as a significant variable is an important consideration in the evaluation of grant proposals by the NIH.

### 2.3. Estrous Cycle

In females, the stage of the estrous cycle must also be considered. In BALB/cBYJ mice, the stage of the estrous cycle significantly affected performance in the open field, tail flick, and tail suspension tasks. For example, mice in estrous had significantly higher anxiety-like behavior than mice in proestrus and metestrus; furthermore, mice in proestrus demonstrated greater sensitivity to pain than in the other phases of the estrous cycle [12]. In C57BL/6J mice, only tail suspension was affected; mice in metestrus showed the highest levels of this depressive-like behavior, whereas tests such as open field, rotarod, startle reflex, pre-pulse inhibition, tail flick, and hot plate remained stable [12]. The stage of the estrous cycle was found to affect spatial memory in C57BL/6 females such that performance was worse in the estrous phase [13]. On a test of social recognition memory, only mice (C57BL/6 and 129 mix) in proestrus demonstrated long-term (24 h) recognition of familiar vs. unfamiliar mice [14]. Interestingly, in female rodents, isolated from males, the estrous cycle can be suppressed or prolonged [15]. This effect, known as the Lee–Boot effect, may decrease variability in behavior studies of females, but it also may complicate an analysis of effects of estrous cycle phases by skewing results so that one stage is represented primarily. If possible, the stage of the estrus cycle at testing should be controlled and documented.

### 2.4. Sex of Siblings

The sex of the siblings can affect behavior of a test mouse. In a study conducted in mice, litters were reduced to six pups with different makeups at birth: 6 females; 1 male, 5 females; 3 males, 3 females; 1 male, 5 females; or 6 males. Testing was performed on postnatal days 18 and 21. The female-skewed litters demonstrated more social play, while the male-skewed litters demonstrated more solitary play. The balanced litters showed increased exploration compared with the skewed litters [16]. Whereas it may be difficult to control for sex balance in a litter, it is important to record and consider its influence.

### 2.5. Sex of Siblings and Location In Utero

Embryos secrete sex hormones during in utero development, suggesting that levels of testosterone/estradiol may vary according to the position in utero. There is evidence that this can mediate postnatal physiology and behavior [17]. When competing for limited food resources, females surrounded by females in utero outcompeted females surrounded in utero by males. Females surrounded by males in utero demonstrated more aggressive behavior as adults than females surrounded by other females in utero [18]. Females surrounded by other females in utero were approached more and were more sexually receptive to male mice [19]. Males surrounded by other males in utero were more parental toward neonatal mice, whereas males surrounded by females in utero committed infanticide more frequently [20]. Although in utero location is not normally known in the context of a typical experiment (it is typically determined by performing caesarian sections in the rodent), it is important to be aware that it may influence behaviors. Furthermore, in experiments in which in utero location is known, recording this variable may be useful in data interpretation.

### 2.6. Genotypes of Siblings

Behavior testing is often performed to determine if there is a behavioral phenotype associated with a particular genotype, but the genotype of the siblings can also affect behavior and the genotype of the offspring can influence maternal behavior which can also influence offspring behavior [21]. For example, male mice with a deletion in Neuroligin-3 (Nlgn3 KO) have deficits in sociability, but wild-type (WT) mice raised together with Nlgn3 KO littermates also displayed deficits in sociability. Re-expression of Nlgn3 in the KO mice normalized behavior in both Nlgn3 KO mice as well as their WT littermates [22]. Littermate-controlled studies may reduce variability due to other aspects such as maternal care and environment (discussed below).

### 2.7. Maternal Care

It is well known that differences in maternal care can alter behavior [23]. In a study of C57BL/6J mice, female offspring raised by mothers showing lower maternal care (determined by pup-licking) showed increased anxiety-like behavior, decreased activity, greater reactivity to a stressor, and reduced pre-pulse inhibition. Male offspring raised by mothers showing lower maternal care had reduced reactivity to a stressor, but were otherwise unaffected by maternal care in the measures examined [24]. Rats that experienced high levels of maternal care showed enhanced spatial learning and memory [25]. Maternal care can affect the subsequent quality of maternal care delivered by the offspring to their pups, thus establishing a transgenerational effect [26], and any notable differences in maternal care should be noted in studies.

## 3. Housing/Husbandry

### 3.1. Marking/Identification of Animals

It is necessary to mark animals for identification purposes, but even this may influence subsequent behavior. Rat pups that were tail-marked for identification had reduced anxiety-like behavior, but paradoxically, as adults, had increased chromodacryorrhea (red tears) in response to handling compared to those that had no markings. In a follow-up study, rats avoided the odor released from the tail-marker pen [27]. These results suggest that even procedures that appear trivial may influence long-term behavior in animals.

Another common method for marking animals is tail clipping, in which animals are often exposed to light isoflurane anesthesia. Exposure to isoflurane alters signaling pathways and results in antidepressant-like behaviors in mice, even six days after administration [28]. Marking/identification of animals should remain consistent throughout a study. If possible, use of sedatives should be reconsidered.

### 3.2. Diet

For some behavior testing, it is necessary to perform overnight fasting. A study of C57BL6/NTac mice revealed that sixteen hours of fasting affected both behavior and physiology, but effects differed between males and females. In females, fasting resulted in elevated heart rate and increased locomotor activity that persisted for 8 and 11 h, respectively. In males, these responses were also observed, but effects were of lower magnitude and only persisted for two hours [29].

Many studies have shown a beneficial effect of caloric restriction on longevity. These effects are mediated by the species and strain, with rats showing the greatest benefit of caloric restriction [30]. Intermittent caloric restriction has been shown to improve the behavioral deficits of a mouse model of Alzheimer’s disease [31]. In the absence of caloric restriction, the pathology and cognitive deficits of a mouse model of Alzheimer’s disease were reversed by treatment with a ghrelin agonist to induce hunger [32] suggesting that hunger, rather than reduced calories, may drive these behavioral results.

The composition of the diet can also affect behavior. A high fat diet has been shown to result in cognitive impairments in C57BL/6 mice [33], and interestingly it was found that learning impairments were induced in male but not in female mice [34]. A high fat diet disrupts circadian rhythms in mice [35]. However, timing the feeding rather than providing the diet ad libitum, even with a high-fat diet, improved circadian rhythm [36]. Timed feeding of a high-fat diet can also improve circadian rhythm in mice with a disrupted circadian clock [37]. In addition, making mice work for their food by wheel running with a low return rate (food-restriction) caused the mice to spontaneously shift their activity phase from night to daytime [38].

The type of diet, the amount of food, and the timing of administration are critical factors in determining an animal’s behavior (sometimes in a sex-specific manner) and should be kept consistent in a study and reported for the sake of reproducibility. It is also interesting to think that small changes in diet may also lead to cognitive enhancements and may be therapeutic in certain disease models.

### 3.3. Housing Density

Housing density can have profound effects on behavior (reviewed [39]). Singly housed mice had reduced corticosterone levels [40], increased activity [41,42], altered anxiety-like behavior [41,43], reduced immobility in the forced swim test [41], impaired performance on various memory tasks [41,42], and impaired social recognition [44]. The effects on anxiety-like behavior were strain- and test-dependent [41,43]. Male DBA and C57BL/6J mice had reduced anxiety-like behavior on the elevated plus maze, but increased anxiety-like behavior in the light/dark and hyponeophagia tests [41]. Male Swiss albino mice had increased anxiety-like behavior on the elevated plus maze [43]. A meta-review showed that social isolation had only a modest influence on anxiety-related defense behavior [45].

Effects of group housing appear to depend on the level of crowding. C57BL/6 mice housed in crowded conditions (8 mice per 18 × 28 cm cage), showed more social avoidance [46]. Even in environments with a normal density of mice (up to five mice per cage), social hierarchy can affect individual behavior particularly in males. Anxiety-like behavior of male mice in the elevated plus maze is affected by the social dominance hierarchy of group-housed animals (up to six mice per 45 × 28 × 13 cm cage [43]) (18 × 17 × 30 cm cage [47]) and more subordinate animals also displayed depressive-like behaviors (as measured by reduced activity in the open-field and increased immobility time in the forced swim test) [47].

Behavioral results are clearly affected by the housing situation. To reduce variability, we recommend keeping the number of mice housed together consistent throughout the study for all animals. Effects of social hierarchy should also be considered.

### 3.4. Neighbors

In the past, it was common practice to house mice and rats in the same room. To mice, rats are predatory animals and exposure to rats can elicit defensive behaviors in multiple strains (more severe in BALB/c and C57BL/6 compared to CD-1 and Swiss-Webster) [48]. As a result, most animal facilities now house rats and mice in separate rooms and limit personnel working with rats from proceeding directly to working with mice. Whereas acute exposure to rats did not affect many behaviors in adult male CD-1 mice, chronic exposure reduced sucrose preference and resulted in increased anxiety-like behavior on the elevated plus maze (EPM) [49]. Adult C57BL/6J mice repeatedly exposed to rats had decreased home-cage exploratory behavior in the active phase, decreased sucrose preference, and increased corticosterone compared to control mice [50]. A study examined groups of male mice housed in a room with and without rats. There were no differences between groups in behavior in the open-field or light-dark box, but mice exposed to rats were more aggressive and more attractive to females. Moreover, naïve mice avoided rat urine, whereas mice housed in the same room with rats investigated it more readily. suggesting habituation to the presence of rats [51]. Housing both species in the same room did not affect weight gain or reproductive behavior in mice [52]. To be safe, as has already been adapted by most facilities, housing mice and rats in the same room should be avoided. If this is not possible, then the co-housing should be noted and kept consistent throughout the study. This is also a consideration for those using behavior testing equipment shared by both species. This equipment should be thoroughly cleaned (even more than normal) to avoid exposure to the other species.

### 3.5. Humidity

Mice of both sexes tested in higher humidity environments tended to show a lower pain threshold as seen in the decreased tail withdrawal latency in response to hot water [10]. It is unclear why humidity would influence this behavior, but it does correspond with many human patients with arthritis complaining of increased pain in high humidity environments [53]. Jackson Laboratory currently recommends humidity levels between 40–60%, though this is a wide range and even variation in this range may affect behavior. Jackson Laboratory also recommends facility temperature of 65–75 °C. It would not be surprising if temperature changes also affected behavior, but effects of temperature on behavior have not been reported. In most animal facilities, temperature is tightly controlled, but humidity less so. Both measures should be documented in papers, and issues with temperature and humidity control should be reported. Animals subjected to significant changes in these variables should be removed from the study.

### 3.6. Cage Changes

In standard animal facilities, it is common to move animals into a new cage with clean bedding either weekly or biweekly. This too can be stressful and influence behavior. Cage changes increased the heart rate, blood pressure, and locomotion in both male and female mice; females had a more prolonged response than males (105 min vs. 75 min). The frequency of cage changes did not alter this response [29], so it appears that mice do not habituate to the effects of cage-changing. Rats increase activity, for several hours, after cage changes [54]. Changing the cage of a mouse significantly decreases its sleep time during that phase [55]. These are important considerations for the timing of behavior testing, and it is advisable to avoid testing on days in which cage-changing occurs.

### 3.7. Bedding

Even the depth of bedding and/or the type of bedding can influence behavior. In a study of golden hamsters, animals were provided with either 10, 40, or 80 cm of wood shaving bedding. The animals with 10 cm of bedding exhibited more wheel activity and wire-gnawing behavior suggesting increased anxiety [56]. Mice housed in shallow bedding displayed more nest building activity than mice in deeper bedding conditions. In female mice, deeper bedding reduced corticosterone levels. These experiments were done in both BALB/c and C57BL/6 mice, and the results in both strains were comparable [57]. CD1 mice were housed with either wood chips or pulp chips (made from paper cellulose) for bedding. Males, but not females, with pulp bedding had significantly better performance on the water T-maze than males with the wood chip bedding. Bedding did not affect female performance. Males with wood chip bedding were significantly less active than males with pulp chips [58]. The exact reason for this is unknown. Pulp chip bedding is more absorbent than wood chip bedding and free from contaminants often present in wood chips. The type of bedding used in a study should remain consistent and be recorded for reproducibility.

### 3.8. Environmental Enrichment

Results of studies of the effects of environmental enrichment on behavior are conflicting [59,60]. One study suggests that the effects of environmental enrichment on behavior is negligible in C57BL/6NTac and DBA/2NCrl mice [61], whereas many other studies report that environmental enrichment can affect mouse behavior [62]. Some behavioral differences in response to environmental enrichment (in Male albino Swiss mice) occurred depending on the time of day (active or inactive phase) of testing [63]. The response to environmental enrichment also varies with sex of the mouse, the form of enrichment used, and the testing variable being considered [64]. The effects of environmental enrichment may also depend on genotype. Differing responses to enrichment were seen in wild-type (WT) and neuropeptide Y knockout mice [65]. Environmental enrichment has been suggested to rescue behavioral deficits seen in the valproic acid mouse model of autism [66], the µ-opioid receptor gene knockout mouse [67], a mouse model with copy number variations [68], and a mouse model of fragile X syndrome [69].

In a study of female C57BL/6N mice (genetically identical), animals were all reared together in one large enriched environment and observed over 3 months for exploratory activity (a readout for behavioral development). Greater individual differences in exploratory activity were noted at the end of the study and were correlated with adult hippocampal neurogenesis (a readout for brain development and plasticity). These results suggest that experience can drive changes in brain plasticity resulting in behavior changes that are independent of genetics [70].

Somewhat similar to this, introducing males to new mice and a new arena resulted in significant changes in many behavioral readouts, especially with regard to activity and aggression [71]. The level of environmental enrichment should be kept constant throughout an experiment to limit variability, and details of environmental enrichment should be included in published studies to allow for reproducibility and interpretation.

### 3.9. Housing Lighting

Both the level of light during testing (discussed later) and during housing can affect behavior. Dim light sources at night can be a major consideration in behavior testing and often difficult to avoid. Light from computers or even from cracks in doors or windows in the housing room can affect mouse behavior. Male C57BL/6J mice chronically exposed to low level light (5 lux) at night and subsequently tested, exhibited no significant differences from controls in total activity but a decreased percent time in the center of the open field suggesting increased anxiety-like behavior [72]. These mice also had significantly reduced preference for sucrose water but no differences in the tail suspension test [72]. Even during the day, position on the cage rack can affect the light level experienced by the mice. The top of the rack system is more exposed to the overhead light and mice housed there often receive 20–80× more light than mice housed at the bottom of the rack. Furthermore, position on the rack can have a differential effect on retinal pathology in BALB/c mice [73] which can clearly also affect behavioral outcomes. This would likely be a difficult factor to control, but grouping animals in a study in either portion of the apparatus would help to avoid this potential confounding factor.

### 3.10. Noise

Noise in the housing/laboratory environment can have profound effects on mice. The noise level can be influenced by changing stations in the housing room that increase the background noise of the room by about 10 dB, heavy personnel traffic, and racks that house a large number of mice [74]. Given that rats and mice are nocturnal, increased noise during the day by personnel working in the facility would likely affect the animals during a time when they are normally sleeping. A review by Rabat [75] highlights the disruption of slow wave and REM sleep by environmental noise and that chronic exposure to noise can result in permanent reduction and fragmentation of sleep and increased corticosterone [75]. Chronically reduced sleep during development has been shown to result in long-lasting behavioral changes in activity, anxiety, and social behavior of C57BL/6J mice [5]. If animals are in a high traffic area, it should be noted. Furthermore, changing to a reverse light/dark schedule might help to alleviate these issues.

## 4. Experiment

### 4.1. Site of Testing

The testing environment can also affect behavior. One study examined the effects of testing in three different laboratories, keeping as many variables as possible the same [76]. Despite the attempt to keep all factors equivalent among sites, the study reports large effects of the site on almost all the tests (open field test, plus maze, water maze, alcohol preference, and response to cocaine). Of course, the experimenter was different across the sites [76]. In another study, ethanol preference and open field activity were stable across different laboratories and across experiments conducted over a 40-year interval. In contrast, anxiety-related behaviors differed across laboratories and even when the same laboratory moved to a different location [77], demonstrating that these behaviors are highly sensitive to changes in location. Different locations might elicit different responses based on noise, light, or even objects in view that may seem threatening or distracting. Testing for an individual study should remain consistent.

### 4.2. Lighting during Testing

In addition to lighting in the housing environment affecting behavior, lighting in the testing environment may also modulate behavior. On the Morris Water Maze, BALB/c mice exhibited poorer performance and higher levels of thigmotaxis in bright light compared with dim light; they also had elevated corticosterone levels [78]. Mice exhibited less thigmotaxis in low light compared with bright light in both closed field testing (similar to an open field with black walls) [79] and in the open field test [80]. It is not surprising that nocturnal animals, like mice, are averse to bright light and that bright light induces stress responses. Lighting conditions for a study should be reported and remain consistent for all animals in the study.

### 4.3. Experimenter

The person conducting testing may also influence rodent behavior. Effects could be due to differences in scent, demeanor, handling techniques. In the presence of a male or a female observer, the outward pain response to injection of an inflammatory agent in both male and female mice was lower in the presence of a male observer. This effect was mimicked by a female observer wearing a t-shirt previously worn by a male. In a separate experiment, plasma corticosterone levels were found to be higher in the presence of a male observer or in the presence of a female observer wearing a t-shirt previously worn by a male. The increase in corticosterone levels is equivalent to a 15 min restraint or a 3 min forced swim test. In accord with this increase in stress, mice showed increased anxiety as measured by increased thigmotaxis in the open field [81]. Both of these results suggest that it is a male scent that influences the behavior. Another study examining the response of different mouse strains to rat odors before and after an ethanol injection was conducted by two different experimenters (one male and one female). Housing of test mice with or without rats, sex of the test mouse, strain of the test mouse, and the experimenter were considered as sources of variability. The greatest source of variability was the experimenter. This suggests, but does not prove, an effect of the sex of the experimenter [82]. There may have been other differences between the two experimenters. In a computational analysis of a large data set of tail-withdrawal latencies following submersion in hot water, the largest influencer among environmental contributors was experimenter, but the effect was not due to age or sex of the experimenter [10]. It was hypothesized that the difference might be due to differences in handling of the animals, although there could also be other unknown factors influencing the variability.

Familiarization with the experimenter can also influence rodent behavior. In a study of anxiety-like behavior in the EPM, rats familiar with the experimenter demonstrated more consistent behavior than rats unfamiliar with the experimenter [83].

Even the presence or absence of an observer in the testing room can influence behavior. In a study of social behavior in C57BL/6J mice, mice were tested in the three-chambered apparatus. Mice were either observed directly or videorecorded. Test mice spent less time sniffing in the absence of an observer. Moreover, the typical preference for social novelty was observed only when the experimenter was present [84].

To reduce variability, it is advisable that the experimenter stay consistent if possible, or, at minimum, the sex of the experimenter stay consistent. Additionally, it is advised that all the animals become familiar with the experimenter prior to testing.

### 4.4. Handling

Rodent behavior is clearly influenced by pretest handling. C57BL/6J male mice handled in either a gentle or an aggressive manner over a two-week period were subsequently tested on the forced swim test; gently handled mice had less immobility compared to aggressively handled mice [85]. In another study, performance was compared between mice handled by the tail or by means of an acrylic tube. Mice were compared on their ability to discriminate between two samples of urine. Mice handled via the acrylic tube performed well on the task [86], whereas mice handled by the tail did not explore the samples. In a study of the effects of pretest handling, Sprague Dawley (outbred) and PVG/OlaHsd (inbred) rats showed an anxiolytic response to pretest handling resulting in increased activity in the open field in both strains [87]. For the best results, we suggest pre-handling rats before testing, and handling mice by means of an acrylic tube. Regardless of method, it should be consistent throughout the study, and it should be reported.

### 4.5. Order/Spacing of Behavior Testing

The number of tests, the order of tests, and the number of days between tests might also influence behavior. Several studies in mice have addressed these issues. To test effects of multiple vs. single behavior tests on performance, male C57BL/6J were subjected to either a battery of behavior tests (spaced out by one week) or a single test. Mice undergoing the battery of behavior tests were less active in the open field, showed increased anxiety-like behavior in the light/dark test, had better performance on the rotarod test, and showed increased pain response (decreased withdrawal latency) on the hot-plate test. Pre-pulse inhibition and conditioned fear were not significantly affected [88]. To test the effects of the order of testing, male C57BL/6J and 129 mice were run on a battery of tests varying the order. Performance on the light/dark test, acoustic startle (only in C57BL/6J mice), and pre-pulse inhibition (only in C57BL/6J mice) were affected by the change in order, but performance on open field and fear conditioning were not [88]. The spacing between behavior testing is another consideration for experimental design. In a study of three strains of male and two strains of female mice, performance was compared on a behavior test battery with either one week or 1–2 days between tests. The only difference found was in the open field in which C57BL/6J males traveled significantly more distance in the shorter spacing design than those on the one-week spacing design. Differences in behavior between these designs were not evident in the light/dark, rotarod, pre-pulse inhibition, and acoustic startle tests [89]. Order and spacing of testing should be consistent and should be reported.

If testing a group of rodents in a single cage, the order of animals tested can also influence behavior. In a study of mice, the first mouse tested tended to have a decreased pain response as seen by higher latencies on a tail-withdrawal task compared to subsequently tested cage mates [10]. In view of the influence of these factors on outcome, it is important to take them into consideration when designing studies and to elaborate details when reporting results.

### 4.6. Drug Delivery

In studies of drug effects on behavior, the mode of drug delivery and the composition of the vehicle can cause pain and stress and consequently affect performance. Injections themselves can alter behavior. For example, following an intraperitoneal injection (IP) of saline, male Swiss mice displayed more anxiety-like behavior on the EPM than noninjected controls [90]. The vehicle for IP drug injections can be irritating and consequently have effects on some behaviors. In studies of male mice (C2DF1), IP injections of Tween-20, Tween-80, DMSO, and ethanol resulted in concentration-dependent decreases in locomotor activity, whereas vegetable oil did not [91]. It is important to understand that a “vehicle-control” may not behave in the same manner as a “control.” The vehicle-control itself may influence behavioral readouts in such studies. These findings are critical in the planning of preclinical treatment studies. Alternative less stressful strategies for drug delivery (such as dietary administration for chronic drug treatments) should be considered.

### 4.7. Time of Day

The time of day of testing can have a role in performance on many behaviors. This may be related to the active (lights off)/inactive (lights on) phases of the circadian cycle in these nocturnal animals or the circadian rhythm of circulating corticosterone levels with levels increasing during the day and decreasing during the night, and there may be other factors. In a study of home cage activity in rats housed with lights on during the day, activity was shown to be higher in the morning compared to the afternoon [54]. Performance of rats on the forced swim test was compared during the active (lights off) and inactive (lights on) phases of the circadian cycle. Results demonstrate that rats engaged in significantly fewer attempts to escape in the active phase [92]. Chronic mild stress is a method to induce depressive-like behaviors in mice. Wistar rats received intermittent chronic mild stress for 6 weeks, such as restricted space for 2 h, tilted cage for 4 h, wet bedding for 8 h, 12 h of food or water deprivation followed by inaccessible food or water for 1 h delivered during the inactive or active periods. Only rats that received the stressors during the inactive phase showed depressive and anxiety-like behavior (as determined by the sucrose preference test, forced swim test, and elevated plus maze), [93]. In mice, the withdrawal threshold to von Frey hairs applied to a hind paw was higher during the inactive phase [94]. For some tests, the ability to discriminate differences in performance among mouse strains depends on the phase of testing. For example, strain differences in performance on open field and rotarod tests were seen only in the active phase, whereas differences in performance on the tail-flick test were seen only in the inactive phase [95]. In contrast, social behavior was unaffected by the circadian phase (reviewed [96]). For the most directly translatable results, studies in the active phase would be best. Nevertheless, noting the time of day for studies and keeping this window fairly narrow is critical (i.e., it is probably better to conduct testing in a group of animals across two days than to test the same group of animals in one very long session).

### 4.8. Season

A previous review has examined the effects of season on tests of rodent behavior. Overall, season does not seem to have a large effect on many behavior tests (such as activity, learning, and sexual behavior) [97], but some may be marginally affected by the season including anxiety-like behavior [97], depressive-like behaviors [97], and response to pain [10,97]. Ideally, studies should be done on a concise timeline. If this is not possible, then ensuring that all groups are equally represented across time is essential (i.e., do not test all controls first and then test experimental animals).

### 4.9. Natural Disasters

Although not a common occurrence, the aftermath of natural disasters can affect the behavior of animals. In Japan, C57BL/6J mice studied before and after a major earthquake showed significantly increased food intake without a subsequent increase in body weight after the earthquake. Mice after the earthquake also demonstrated significantly higher anxiety-like behavior, enhanced performance on the Morris water maze, and significantly higher serum corticosterone levels [98]. These data may also be relevant to other natural occurrences such as violent thunderstorms and hurricanes. Even the COVID-19 related restrictions may affect rodent behavior possibly due the reduced traffic of personnel in animal facilities. These events are clearly impossible to “control” for, nevertheless, they should be recorded. In particular, with respect to COVID-19 pandemic restrictions, it would be best to state at what point the testing was conducted. If some of the data were collected before the pandemic, and then some testing during, this should be noted and analyzed as a potential confound.

## 5. Gene × Environment Interactions

Although any number of the factors reviewed here can influence behavior, it is the combination of these factors that likely contribute to differences in behavior. Several large studies of mice have examined gene × environment interactions in behavioral assessments. A computational analysis of behavioral results from 8034 adult mice of different strains/substrains was carried out on tail-withdrawal latencies in response to submersion of the tail in hot water [10]. Genotype (strain/substrain) accounted for 27% of the variability, environment for 45% of the variability, and the gene × environment interaction accounted for an additional 15% of the variability. In another mouse study of 88 phenotypes (behavioral and physiological), genetic variation accounted for less than 50% heritability, and influence of environment accounted for usually less than 10%. The largest contributor was the interaction between gene and environment which sometimes exceeded 50% [99]. Accordingly, analyzing the genetic and environmental contributions separately will not give a complete picture of variability in a study. Genetic × environmental interactions should always be considered.

## 6. Summary

The strains used in each study should be consistent, as well as explicitly reported (including substrain), for reproducibility. It is important to study males and females separately to yield complete and accurate results. If possible, for females, the stage of the estrus cycle at testing should be controlled and documented. It is important to record the sex balance of a litter. If known, it may be useful to record in utero location. Litter information should be recorded, like genotypes and which animals were litter-mate controls. Any notable differences in maternal care should be noted in studies.

Marking/identification of animals should remain consistent throughout a study. If possible, the use of sedatives should be reconsidered. Diet (type and amount) and timing of food delivery should be kept consistent and reported for the study; in addition, if the amount of food consumed is known, it should be recorded. The number of mice housed together should be kept consistent throughout the study for all animals, and the effects of social hierarchy should also be considered. Whether mice and rats are housed in the same room should be consistent and recorded, and this should also pertain to equipment used for both species. Humidity and temperature should be reported and kept consistent throughout the study. If possible, it is advisable to avoid testing on days in which cage-changing occurs. The type of bedding used in a study should remain consistent and be reported for reproducibility. The level of environmental enrichment should be kept constant throughout an experiment to limit variability, and details of environmental enrichment should be included in published studies to allow for reproducibility and interpretation. It would be ideal to house animals in similar portions of a cage rack to keep the lighting in the housing condition consistent. If animals are in a high traffic area, it should be noted. Furthermore, changing to an inverted light/dark schedule might alleviate issues with high noise levels affecting sleep time.

The testing site for an individual study should remain consistent. Lighting conditions for testing should be reported and remain consistent for all animals in the study. To reduce variability, it is advisable that the experimenter stay consistent if possible, or, at minimum, the sex of the experimenter stay consistent. Additionally, it is advised that all the animals become familiar with the experimenter prior to testing. For the best results, it may be best to handle mice with an acrylic tube and to pre-handle rats before the study. Regardless of handling method, it should be consistent throughout the study and be reported. Order and spacing of testing should be consistent and be reported. Even order of testing mice in a cage should be considered in the results and reported. Less stressful strategies for drug delivery should be considered. Studies in the active phase might be the most translationally relevant, but time of day should always be noted and kept within a narrow window. Studies should be limited to one season and reported when they were performed. Natural disasters or events immediately preceding or even during studies should be noted and considered as confounds.

## 7. Conclusions

The goal of this review was to inform the reader of various factors that influence behavior in rodents, and therefore, things to consider when designing and executing studies. Some are likely to have more of an influence than others. The easiest to control are likely strain, sex, identification methods, diet, whether or not mice and rats are housed in the same room, bedding, environmental enrichment, the site of testing, lighting of testing, the order of testing, drug delivery methods, and the time of day. In addition to issues that can lead to effects on behavioral outputs in animals, careful consideration of power, sample size, and statistical treatments [100] should be given. The main point of this review is to strongly encourage researchers to include as much detail in the methodology as possible to aid in interpretation and reproducibility of the results. Moreover, interpretation of results must be done cautiously, as many factors can contribute to behavioral differences. It may be difficult to consider results from many different labs together, as differences will likely abound. Clearly, it is a herculean task to control for all the influences on behavior, but being aware of them should aid in the design and execution of a study.

## Data Availability

Data sharing not applicable.

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
