# Peer review of "Behavior Testing in Rodents: Highlighting Potential Confounds Affecting Variability and Reproducibility"

_brainsci, 2021, doi:10.3390/brainsci11040522_

Round 1

Reviewer 1 Report

Excellent well written and comprehensive review article. I would like a summary of recommendations (which were embedded within each section, and thus not easily accessible) at the end in the discussion section, but otherwise no changes recommended.

Author Response

Thank you for your time and review.  We have added a summary section with a listing of all recommendations.

Reviewer 2 Report

Rachel et al. have reviewed and highlighted various potential confounding factors which might be causal in behavioral variability and reproducibility. Many confounding factors, which authors have listed, are the reasons for the failure of preclinical studies to human or industrial trials. I agree with the authors that it is of utmost importance to be aware of and implement conditions to minimize or standardize these variabilities following what has been published.

Major points:

  1. Authors have stressed reproducibility but started with a factor of "strain" [section 2.1, line 53-58] and wrote 5 liners with 2 references or
  2.  A short paragraph on the sex of siblings [Section 2.4, line 82-89] with just one reference.
  3. There are many more such examples, like Section 2.6, 2.8, 3.1, 3.9, 4.1, 4.7, 4.9, etc., just to name a few where either one or two references have been used to highlight the confounding factor. However, this under-reporting of references undermined their review as the whole point is to tell the reader about the "variabilities" obtained in multiple studies with such and such factors involved.
  4. The authors have completely skipped one major factor in rodent research, which is a prime reason for the large failure of rodent preclinical research into the clinic. That is the sample size of behavioral research. Please write a section on that, and you may refer to."

    "Power failure: why small sample size undermines the reliability of neuroscience by Katherine S. Button et al., Nature review neuroscience, 2013" and much other such paper to write this section.

  5. Some papers have analyzed confounding factors like gender and analyzed their effects on brain-related behaviors like "Concordance and incongruence in preclinical anxiety models: Systematic review and meta-analyses, Farhan Mohammad et al. Neuroscience & Biobehavioral Reviews, 2016". Please include inferences from this paper in your review.

Minor:

  1. There are misplacements of. In some of the sections, for example, 2.1. Strains: It should be 2.1. Strains: there are many such issues. Please revise the manuscript to fix these typos.
  2. Section 2.2 could be labeled as Gender differences instead of Sex
  3. Sections "Experience" and "housing density" could be merged into one section.
  4. There are two sections by the title "lighting." Please change these subheadings according to the section—for example, housing lighting and testing lighting.
  5. The conclusion is really very brief. Please elaborate on the issues, highlight studies that have taken care of these issues and future perspectives. 
  6. I also believe a better title could be (if authors like it|) " Potential confounding factors affecting behavioral variability and reproducibility in rodents I don't know where they have separated confounds from factors?.

."

Author Response

Open Review

English language and style

( ) Extensive editing of English language and style required
( ) Moderate English changes required
(x) English language and style are fine/minor spell check required
( ) I don't feel qualified to judge about the English language and style

Is the work a significant contribution to the field?

Is the work well organized and comprehensively described?

Is the work scientifically sound and not misleading?

Are there appropriate and adequate references to related and previous work?

Is the English used correct and readable?

Comments and Suggestions for Authors

Rachel et al. have reviewed and highlighted various potential confounding factors which might be causal in behavioral variability and reproducibility. Many confounding factors, which authors have listed, are the reasons for the failure of preclinical studies to human or industrial trials. I agree with the authors that it is of utmost importance to be aware of and implement conditions to minimize or standardize these variabilities following what has been published.

Major points:

  1. Authors have stressed reproducibility but started with a factor of "strain" [section 2.1, line 53-58] and wrote 5 liners with 2 references or

The references were well-written review articles that already provided good information on this subject.

  1. A short paragraph on the sex of siblings [Section 2.4, line 82-89] with just one reference.
  2. There are many more such examples, like Section 2.6, 2.8, 3.1, 3.9, 4.1, 4.7, 4.9, etc., just to name a few where either one or two references have been used to highlight the confounding factor. However, this under-reporting of references undermined their review as the whole point is to tell the reader about the "variabilities" obtained in multiple studies with such and such factors involved.

At the outset, we stated that we were not aiming to provide a comprehensive review or an annotated bibliography.  Rather, our aim was to introduce the reader to factors that needed to be considered when designing and interpreting behavior studies.  Some of these effects have been comprehensively reviewed elsewhere and many are not well studied.  In the case of the former we included the reviews as citations and in the case of the latter we cite what was available in the literature.

  1. The authors have completely skipped one major factor in rodent research, which is a prime reason for the large failure of rodent preclinical research into the clinic. That is the sample size of behavioral research. Please write a section on that, and you may refer to."

"Power failure: why small sample size undermines the reliability of neuroscience by Katherine S. Button et al., Nature review neuroscience, 2013" and much other such paper to write this section.

We agree that this issue is important for the failure of rodent research to translate into the clinic.  However, as this is not something that directly affects the behavior of the animal, nor is a factor affecting variability among animals.  As such we considered this issue out of the scope of our review.

  1. Some papers have analyzed confounding factors like gender and analyzed their effects on brain-related behaviors like "Concordance and incongruence in preclinical anxiety models: Systematic review and meta-analyses, Farhan Mohammad et al. Neuroscience & Biobehavioral Reviews, 2016". Please include inferences from this paper in your review.

We have added this reference.

Minor:

  1. There are misplacements of. In some of the sections, for example, 2.1. Strains: It should be 2.1. Strains: there are many such issues. Please revise the manuscript to fix these typos.

We are unsure of what the reviewer is referring to in this instance.

  1. Section 2.2 could be labeled as Gender differences instead of Sex

Mice do not identify with a particular gender, but only have biological sex.

  1. Sections "Experience" and "housing density" could be merged into one section.

As per another reviewer’s comments, we have moved the experience section.

  1. There are two sections by the title "lighting." Please change these subheadings according to the section—for example, housing lighting and testing lighting.

We have changed this.

  1. The conclusion is really very brief. Please elaborate on the issues, highlight studies that have taken care of these issues and future perspectives. 

As per another reviewer’s comments, we have expanded the conclusion section.

  1. I also believe a better title could be (if authors like it|) " Potential confounding factors affecting behavioral variability and reproducibility in rodents I don't know where they have separated confounds from factors?.

We have edited the title.

Reviewer 3 Report

I think that the authors wrote an interesting review with useful consideration for that may help in the standardisation of behavioural procedures. Please find below some specific comments:

Line 54: the sentence do not sound complete

Line 67-69: in this sentence the authors listed a number of behavioural evidence without references, which should be added. Same thing happened in lines 116-119, and 126-128, 128 -132.

The paragraph "Experience" in my opinion is not well explained. I think the title is not appropriated. The authors discussed (briefly) only about enrichment. However, given that there is already a paragraph "Environmental enrichment" I do not understand the difference. May be these two could be merged

Regarding this paragraph may be the authors could consider the following article: Forkosh et al 2019 Nat Neurosci doi.org/10.1038/s41593-019-0516-y. 

line 174: please check whether the sentence is correct

Line 141-144: please add references

Humidity is normally controlled in animal facilities. May be this could be added (together with the normal levels of humidity allowed, and temperature ?)

Lighting paragraph needs references. 

May be the authors could add a comment about influence of noise. What could be the solution? testing with inverted cycle? 

Line 308-317: Is this all part related to the same reference? if yes, please introduce the reference earlier. 

Line 353-359: please add references

Line 364-371: same as before, Is this all part related to the same reference? if yes, please introduce the reference earlier. 

Author Response

Thank you for your time and reviews.  Please see the attachment.

Reviewer 4 Report

In this manuscript, Sare’ and colleagues review a number of factors that can influence behavioral measurements in rodents. Their goal is to increase awareness and emphasize the need for detailed reporting of the methods used. This topic is very relevant for the reproducibility of behavioral and physiological measurements within and across laboratories. The manuscript is well written—I personally learned a lot—and I only have a few minor comments.

  • The main focus of this review is on ‘rodents’. This group includes more that 2000 species, but here the discussion is primarily on mice and, more rarely, on rats. This is very natural considering that other rodent species are not typically used in the laboratory. However, I found the rare references to rats mixed here and there with those of mice somewhat distracting . As a personal preference, and considering that these are different species, I would have preferred a dedicated short-ish section on rats. This is a personal preference that the authors should feel free to disregard.
  • Some sections have clear recommendation and advices, while others do not. I do not think the authors wanted to covertly suggest some sort of “ranking”, but the discontinuous presence of explicit recommendations across sections somehow suggests it. I would clarify.
  • Related to the point above, it seems to me that some factors are more relevant and significant than others in driving physiological and behavioral variability. I would have personally enjoyed a final section with the “essentials”: granted that all reviewed factors can contribute to the variability of various measurements, some factors should be considered as absolutely necessary to report (although not sufficient) for improved reproducibility of the measurements. I think the authors’ opinion on these “essentials” could provide useful guidelines to the reader.
  • I suggest citing Hut et al., PLoS One 2011 on food availability and its impact on the circadian rhythm.

Author Response

Thank you for your time and review.  Please see the attachment.

Round 2

Reviewer 2 Report

Authors response: We agree that this issue is important for the failure of rodent research to translate into the clinic. However, as this is not something that directly affects the behavior of the animal, nor is a factor affecting variability among animals.  As such we considered this issue out of the scope of our review.

My final comments: The authors have modified their review in the line suggested by other reviewers and me. While most comments are adjusted, I am surprised that the authors believe that sample size does not affect the variability and reproducibility. This is in the title of the review. The authors seem not interested in writing a paragraph on the sample size and its importance in understanding variability and reproducibility in behavioral neuroscience. I will suggest they write down few lines in either discussion or conclusion about it and citing relevant publications.

Author Response

Since sample size does not affect the behavior of the animal, we have included a sentence in our conclusion about other important factors affecting variability and translatability of studies as the reviewer suggested.